ecology/behaviour/taxonomy and systematics

periodical organisms, millipedes, train obstructions, mass emergence, life cycles, periodicity

**Author for correspondence:**
Jin Yoshimura
e-mail: yoshimura.jin@shizuoka.ac.jp

†Present address: 4-12-18 Eifuku, Suginami, Tokyo 168-0064, Japan.

# Eight-year periodical outbreaks of the train millipede

Keiko Niijima[1,†], Momoka Nii[2]
and Jin Yoshimura[2,3,4,5,6,7]

[1]Former Tama Forest Science Garden, Forestry and Forest Products Research Institute, Hachioji, Tokyo 193-0843, Japan
[2]Department of Mathematical and Systems Engineering, Shizuoka University, Hamamatsu, Shizuoka 432-8561, Japan
[3]Department of International Health and Medical Anthropology, Institute of Tropical Medicine, Nagasaki University, 1-12-4 Sakamoto, Nagasaki 852-8523, Japan
[4]Department of Biological Science, Tokyo Metropolitan University, Hachioji, Tokyo 192-0397, Japan
[5]University Museum, the University of Tokyo, Bunkyo-ku, Tokyo 113-0033, Japan
[6]Department of Environmental and Forest Biology, State University of New York College of Environmental Science and Forestry, Syracuse, NY 13210, USA
[7]Marine Biosystems Research Center, Chiba University, 1 Uchiura, Kamogawa, Chiba 299-5502, Japan

JY, 0000-0003-1610-1386

Periodical cicadas are the only confirmed periodical animals with long life cycles. In Japan, however, 8-year periodicity had been suggested in a species of train millipedes that had frequently obstructed trains in the central mountainous region of Honshu, Japan. This species was identified as *Parafontaria laminata armigera* Verhoeff (Diplopoda: Xystodesmidae), which is endemic to Japan. We finally confirmed the 8-year periodicity of this millipede using detailed surveys of life histories over 8 years. Seven broods were recognized, with almost no overlaps in their distributions. We also report the historical outbreaks and train obstructions of this millipede during 1920–2016. This is the first confirmed case of periodical non-insect arthropods.

## 1. Introduction

Periodical organisms are unique in having fixed long life cycles (periodicity) and undergoing mass emergence [1]. In plants, many species of bamboos and the genus *Strobilanthes* are known (or mostly suspected) to be periodical mass-flowering species [2,3]. In the animal kingdom, periodical cicadas are the only known and firmly confirmed periodical animals, having 17- and 13-year life cycles, except for species with short cycles of 5–6 years or less (most with cycles of 2 years) [4,5].

In Japan, outbreaks of train millipedes have been reported since 1920 [6,7]. Several outbreaks had been observed in autumn along Chuo Line, Koumi Line and Hokuriku Line; the millipedes are identified as *Fontaria laminata* Attem [6]. At that time, periodicity had never been recognized because of irregular emergence periods and many swarming events in these areas [7]. Later, these outbreaks were found to include several species of millipedes (see below). Furthermore, it had not been known how long the adults that appeared during an outbreak or swarming survive. Note that there are many millipede species in which adults survive several years in their life cycles [8]. The swarming of pre-adult (7th instar) that had occurred sometimes also seemed to be included in the outbreak/swarming records. In addition to these problems, there was an evident error in the outbreak year. Because of all these problems, the periodicity of these millipedes had never been recognized in those days.

Later, the genus *Fontaria* was renamed to *Parafontaria*, and the train millipedes were divided as follows: (i) *Parafontaria laminata armigera* Verhoeff 1936 (Diplopoda: Polydesmida: Xystodesmidae) (*P. l. a.*), (ii) Echizen train millipede: *P. echizenensis* Shinohara 1986 (*P. e.*), (iii) Toyama train millipede: *P. kuhlgatzi* Verhoeff 1937 (*P. k.*), and (iv) related species Obibaba-yasude: *P. laminata laminata* Attems 1909 (*P. l. l.*) [9,10]. The name train millipede is derived from the frequent train obstructions during 1920–1984 caused by *P. l. a.* and *P. e.*.

After the taxonomic corrections of the *P. laminata* group, *P. l. a.* was suggested to be the 8-year periodical millipede [11] because of the following observations. First, the taxonomic corrections revealed that many outbreaks/mass emergences are found to belong to *P. l. a.* and 8-years periodical at each location. Second, a 4-year consecutive survey indicated that *P. l. a.* moults once a year in summer (due to the need for long cold winter for moulting) and becomes adults in the eighth year because millipedes belonging to the order Polydesmida had been known to moult seven times to be adult in the 8th instar [8,12]. Thus, *P. l. a.* was suggested to be an 8-year periodical millipede [11], but these incomplete/partial observations could not confirm the 8-year periodicity in this species. Later, various data further supported the suggested 8-year periodicity in *P. l. a.* [9,13–15], but we could not yet confirm its 8-year periodicity because we have no data on its complete life cycle, nor the analysis of negative data in the years of the expected no emergence.

Here, we report the confirmation of the 8-year periodicity of *P. l. a.* from close tracking observations of field growth schedules from adults to the adults in the next generation in two different localities (different broods). We also gather historical outbreaks and train obstructions of this millipede between 1920 and 2016. We then present the detailed distributions of all observed broods (emergence year groups) in Central Japan. Notably, all these survey data also confirm the 8-year periodicity of this train millipede.

# 2. Material and methods

## 2.1. Study sites and methods for field surveys

We selected two main sites for the close survey for growth schedules: (i) Mt. Yatsu site and (ii) Yanagisawa site (figure 1*a* and electronic supplementary material, table S1). The Mt. Yatsu site is located near the Koumi Line that had been reported for frequent train obstructions (figure 1*b*). In this site, the first author had conducted the observation during 1974–1980 and 1982–1985 (7 and 4 consecutive years respectively), 1994, 2000, 2001, 2008 and 2016 (total 16 years) (figure 2*a*–*c* and electronic supplementary material, table S2). The Yanagisawa site is located in the Chichibu-Tama-Kai National Park east of the Mt. Yatsu site. In this site, the first author had conducted the observation during 1972–1975, 1977–1985, 1987–1988, 1995–1998 (4, 9, 2, 4 consecutive years, respectively), 2005, and 2012 (total 21 years) (figure 2*d*–*e* and electronic supplementary material, table S3). The detailed explanation of each sampling point and sampling procedures are shown in the electronic supplementary material.

### 2.1.1. The Mt. Yatsu site

The Mt. Yatsu site (site 4 in figures 1*a* and 2*a*–*c* and electronic supplementary material, tables S1 and S2) is in the south margin of Yatsugatake-Chushinkogen quasi-national park. The main study points (m1, m2 in figure 2*a*) were located on the north slope of Mt. Meshimori (35°55′58″ N, 138°27′32″ E; 1300 m.a.s.l.) in Nobeyama, Minamimaki Village, Nagano Pref.; regarding vegetation, these points are within a larch plantation (*Larix kaempferi*) (figure 2*c*) that was planted in 1962 and has been managed as the Nagano Prefectural Forest. Weather records were provided by the Nobeyama Office of Tsukuba University

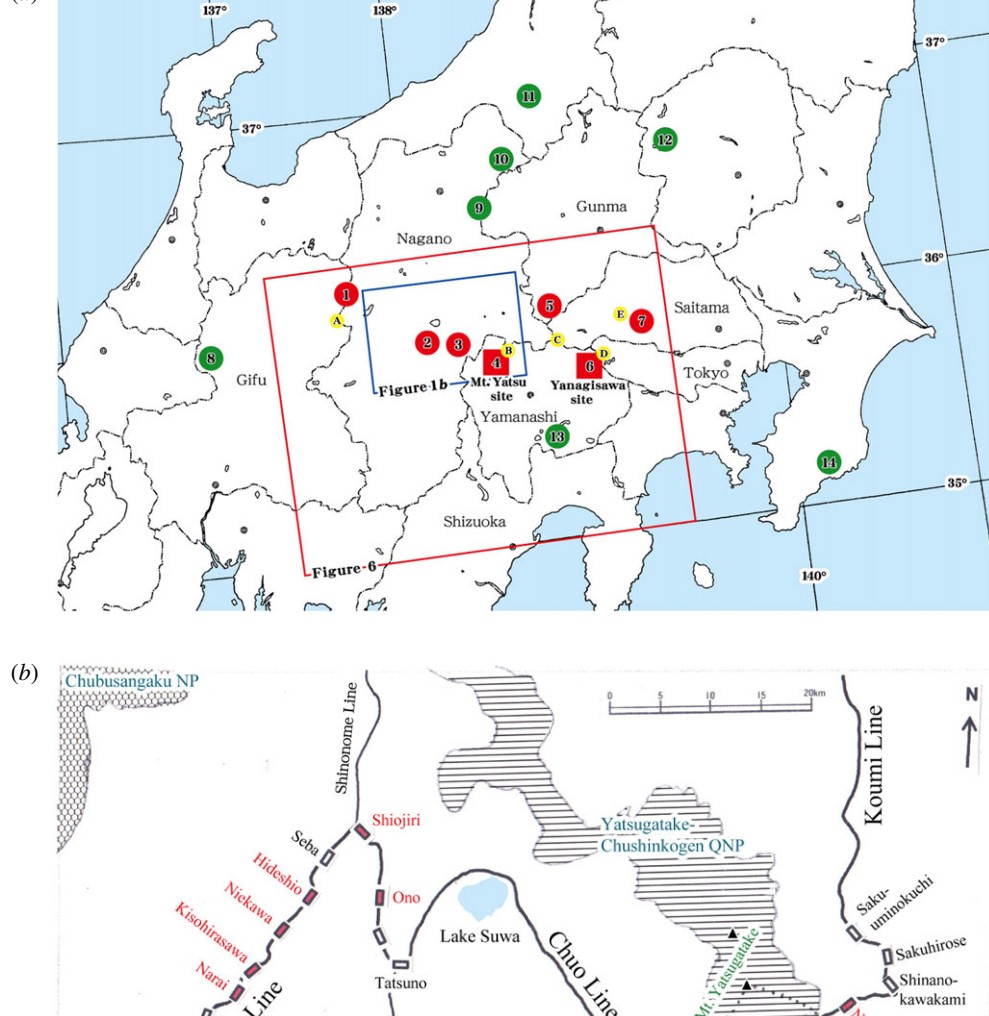

**Figure 1.** The study sites and the sections where the millipede obstructed trains. (a) The study sites. The train millipede, *P. laminata armigera*, was collected at sites 1–7 (red circles or squares). The two main study sites are Mt. Yatsu (site 4) and Yanagisawa (site 6). Closely related species were collected at sites 8–14 (green circles) (electronic supplementary material, table S1). Points A–E (yellow circles) are the meteorological data points (electronic supplementary material, table S4). The two squares indicate the areas shown in the enlarged maps (red lined: figure 6; blue lined: figure 1b). (b) The sections of train obstructions by the millipede on the Chuo Line and Koumi Line (sections between adjacent stations in a red letter). National parks (NP; mesh pattern) and quasi-national parks (QNP; horizontal pattern) are also shown.

(B in figures 1a and 2a), and the mean air temperature and annual rainfall at this point were 6.9°C and 1371 mm, respectively (B in electronic supplementary material, table S4).

The millipedes were surveyed one to five times per year by hand sorting at point m1 in 1974–1980 and at point m2 in 1982–1994 (electronic supplementary material, table S2). One to five quadrates of 25 or 50 cm squares were set and the soil dug out to a depth of 15–20 cm. The soil to a depth of 0–5 cm was dug out, spread on a polyethylene sheet and the millipedes on the sheet were collected using forceps or an aspirator (figure 3a). Then, the same procedure was repeated for 5–10, 10–15 and 15–20 cm depths. Smaller invertebrates were also collected to avoid overlooking the immature stage of the millipede.

Soil samples for soil microarthropods, including nymphal millipedes, were collected using stainless steel tubes (size: 100 cm² wide by 4 cm deep; volume: 400 ml) (electronic supplementary material, table S2).

R. Soc. Open Sci. **8**: 201399

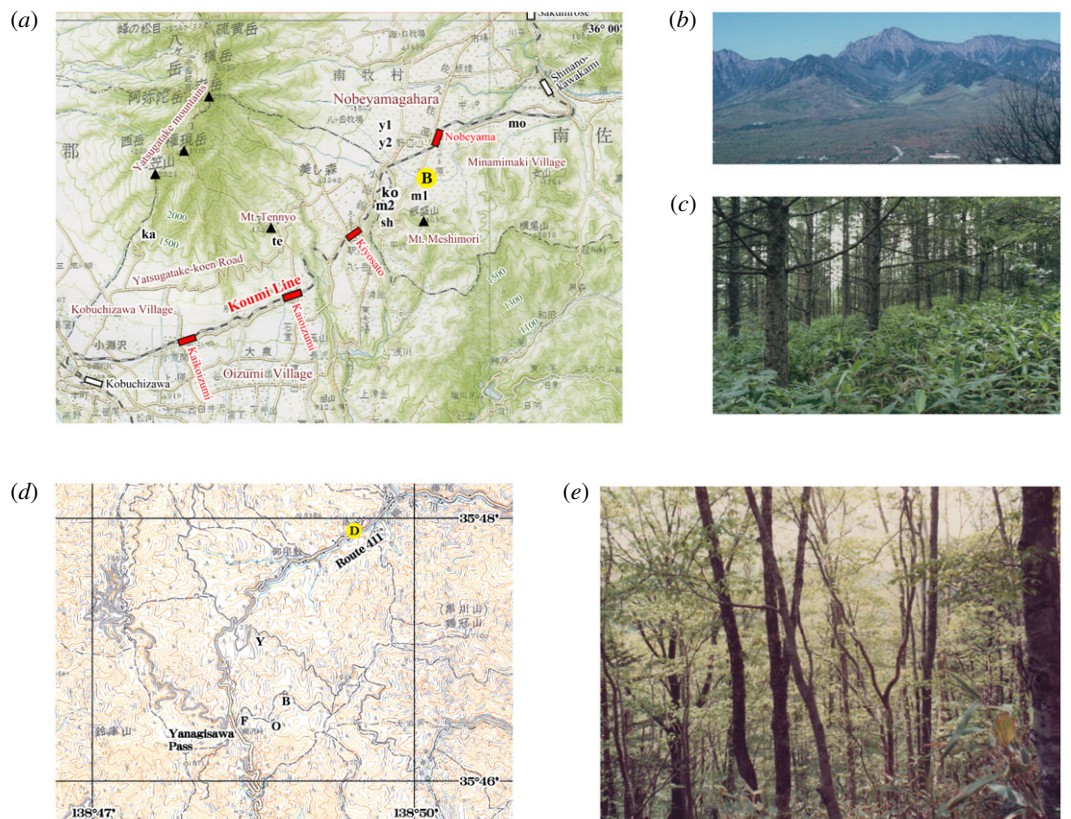

**Figure 2.** The maps and photographs of the study points. (*a*) Study points in the Mt. Yatsu site (m1, m2: main study points; y1, y2: Yatsugatake Experiment Forest of Tsukuba University; B (yellow circle): Nobeyama Office of Tsukuba University for weather records. The other points are explained in the electronic supplementary material. Train obstructions occurred between the stations in red. (*b*) Yatsugatake Mountains (18 October 1984). (*c*) Larch plantation at the point m2 (4 August 1982). (*d*) Study points in the Yanagisawa site (B: beech forest; F: beech with fir forest; O: oak forest; Y: Yokote Forest Road; D (with a yellow circle): Ochiai for weather records). (*e*) The beech forest at the point B (26 June 1983). (*a*) Based on a 1/200 000 topographical map of 'Kofu' and (*d*) based on a 1/50 000 topographical map of 'Taba' published by the Geospatial Information Authority of Japan Institute. All photographs in this article were taken by K. Niijima.

Swarming and/or aggregating millipedes were observed at points m1 and y1 in figure 2*a* in the autumn of 1984 and at point y2 in the autumn of 2000. The number of swarming millipedes was counted inside a wooden frame of $25 \times 25$ cm$^2$ or $50 \times 50$ cm$^2$ on the road or photographs with the frame were taken, and the millipedes were counted later. The collected animals were fixed in 80% alcohol immediately after collection.

### 2.1.2. The Yanagisawa site

The Yanagisawa site (site 6 in figures 1*a* and 2*d*–*e* and electronic supplementary material, tables S1 and S3) is in the water conservation forest of Tokyo Metropolis (area: 21 627 ha). The study points are located near Yanagisawa Pass (35°46′36″ N, 138°49′49″ E; 1500 m.a.s.l.), Koshu City, Yamanashi Pref. The area is covered primarily by natural forests of deciduous broad-leaved trees. The mean air temperature and annual precipitation at Ochiai, Koshu City (D in figures 1*a* and 2*d*) were 8.6°C and 1692 mm, respectively (D in electronic supplementary material, table S4). In this area, many forests are preserved and protected as part of the national park, the water conservation forest and Saitama Prefectural Natural Parks. The density of soil invertebrates at the site was reported by Niijima without further classification beyond Diplopoda [16]. The species and life stage of the collected millipedes were identified in this study.

### 2.1.3. Other study sites

The other study sites (figure 1*a* and electronic supplementary material, table S1) with the sampling date and species of the collected millipedes (*Parafontaria laminata* group) are shown in electronic supplementary material, table S1. Weather records at points A–E in figure 1*a* are shown in electronic supplementary

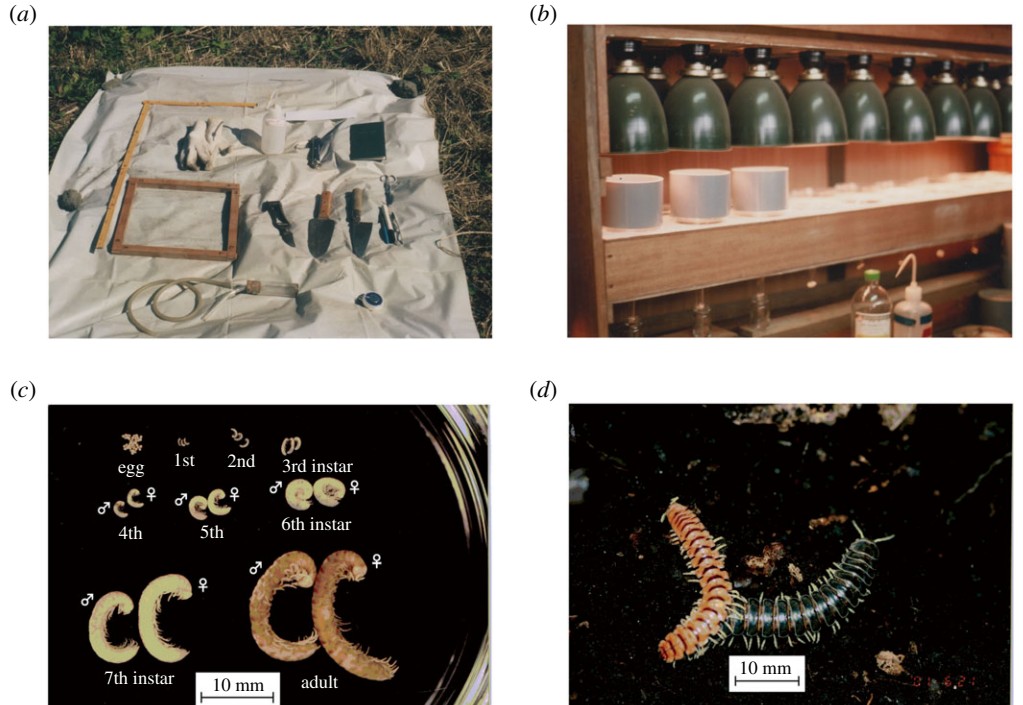

**Figure 3.** Millipede collection and sorting methods. (*a*) Collecting tools for soil macrofauna. (*b*) The Tullgren apparatus for extracting soil micro arthropoda. (*c*) The life stages of *Parafontaria laminata armigera* (also figure 5). (*d*) Adults of *Parafontaria laminata armigera* (left: reddish orange with brown stripes) and *P. tonominea* (right: shiny black). Photos by K. Niijima.

material, table S4. The vegetation, altitude, latitude and longitude of sites 1–7 in figure 1*a*, where *P. l. a.* was collected, are shown in electronic supplementary material, table S8. At sites 1–14 in figure 1*a*, millipedes 0–10 cm deep were collected at 10 points/site.

## 2.2. Laboratory experiments

A collection of nymphal millipedes was carried out as follows. Microarthropods, including nymphal millipedes, were extracted using the Tullgren apparatus (figure 3*b*) [16], which was used for soil samples of 100 or 400 ml, and the Berlese apparatus, which was used for soil samples of $25 \times 25 \times 5$ cm$^3$. The Berlese apparatus was made of a tin plate with a 2 mm mesh sieve, 30 cm in diameter and a 60 W light bulb. The extracted invertebrates were sorted under a dissecting microscope, nymphs of Polydesmida were picked up. Their genus and life stage were identified under a microscope and only *Parafontaria* were counted.

The species identification of *P. l. a.*, which is distinguished from the other three species of train millipedes, is carried out by comparing the male genitalia of specimens [10]. The species identification of previous reports was corrected by Shinohara [17]. We followed his species identification method in this report.

The life stages (instar, stadium) of Polydesmida millipedes were determined by their segment and leg numbers (figures 3*c* and 5*c*), but differences in shape among species are not known [8,12]. The *Parafontaria* spp. collected in the two main study sites were *P. l. a.* and *P. tonominea* Attems, 1899 (figure 3*d*). Adults and the 7th instar of these two species are easily identified: *P. tonominea* is larger in size than *P. l. a.* At these sites, *P. tonominea* (adults and the 7th instar) was extremely scarce (less than 1 out of 10 000 millipedes in photos of swarming/aggregation). Therefore, all nymphal *Parafontaria* sp. (the 1–6 instars) in these sites were treated as *P. l. a.*

## 2.3. Emergence reports and surveys

The researchers of the Prefectural Forest Research Center patrolled the main mountain path at least once or twice a month, except during the winter snow season. Some of them were interested in the outbreak of the millipede and gave us information on the millipede with specimens, photographs and data on the collection locations.

(a)

| year[1] Sep.-Oct. | emergence period | emergence state of the millipede[2] | notes and lines obstructed |
|---|---|---|---|
| 1911 | | | Chuo line inaugurated |
| 1912 | 8 | outbreak | (hearsay of the locals) |
| 1920 | 8 | outbreak | Nie.-Narai (CL) |
| 1928 | 8 | outbreak | Ono-Shio. (CL) |
| 1935 | | | Koumi line inaugurated |
| 1936 | 8 | outbreak | Hide.-Kiso. (CL) |
| 1936 | 8 | outbreak | Kaio.-Nobe. (KL) |
| 1943 | 7 | outbreak | Kaio.-Nobe. (KL)[3] |
| 1943 | 7 | outbreak | Shina.-Saku. (KL)[4] |
| 1944 | 8 | | no record because of |
| 1945 | | | World War II (1941–1945) |
| 1952 | 8 | outbreak | Kiyo.-Nobe. (KL) |
| | | no | |
| 1960 | 8 | outbreak | Kiyo.-Nobe. (KL) |
| | | no | |
| 1968 | 8 | outbreak | Kaiko.-Nobe. (KL) |
| | | no | |
| 1976 | 8 | outbreak | Kaiko.-Nobe. (KL) |
| 1976 | 8 | outbreak | Ono-Shio. (CL) |
| | | no | |
| 1984 | 8 | outbreak | Kaiko.-Nobe. (KL) |
| | | no | |
| 1992 | 8 | mass emergence | |
| | | no | |
| 2000 | 8 | mass emergence | |
| | | no | |
| 2008 | 8 | swarming | |
| | | no | |
| 2016 | 8 | swarming | |

1. Red, blue and yellow years: state of the millipede was recorded, black years: no record.
2. outbreak: millions, mass emergence: tens of thousands, swarming: hundreds, no: the millipede was not found.
3. 7th instar of Brood VI suspected.
4. Brood V (see figure 6).

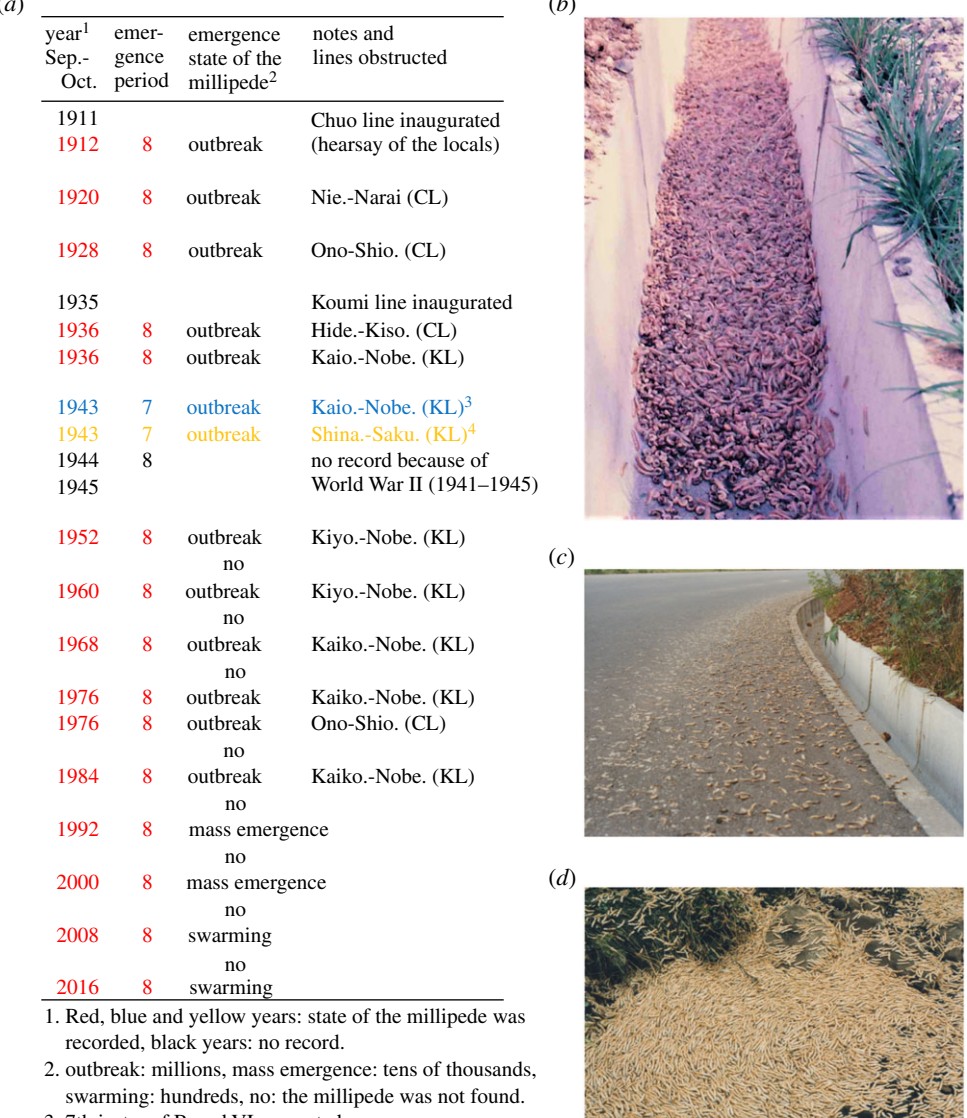

**Figure 4.** The train obstructions by outbreaks of the millipede (electronic supplementary material, table S5). (a) Obstruction of the Chuo Line (CL) and the Koumi Line (KL) occurred in 8-year intervals during 1920–1984, except for in 1944. The full name of the stations and their locations are in figure 1b and electronic supplementary material, table S5. (b) The train millipede aggregated in a road gutter (16 September 1976). (c) Hundreds of train millipedes squashed by cars on a road (24 September 1984). (d) Aggregation of thousands of train millipedes (26 September 1984).

A handbill requesting records or information of the *P. laminata* group has been distributed among forestry researchers and school teachers since 1976, when a large-scale outbreak occurred at the Mt. Yatsu Site, Nagano Pref. [9,11]. A website with a short report on the train millipede has been set up since 1996 [18]. Many people (unknown or acquaintances) provided us with information about the millipedes, sometimes with photographs and/or specimens. We surveyed these points as much as possible.

The records on train obstructions of the Koumi and Chuo Lines by the train millipede (figures 1b and 4a and electronic supplementary material, table S5) were based on the information on the track maintenance records provided by staff members of the Japan National Railway (JNR). The JNR staff checked the railway track condition and wrote the diary of the daily track maintenance record. They compiled the report of train obstruction and made the internal reports of train obstructions in 1936, 1938, 1976 and 1984 (electronic supplementary material, table S5). The current reports are based on these internal reports. We also surveyed the newspapers and other references for the train obstruction by millipedes before 1936. The current report is based on the new taxonomy and corrected data explained in the Introduction [9,10,17,19]. We also added the survey data and observations after these records.

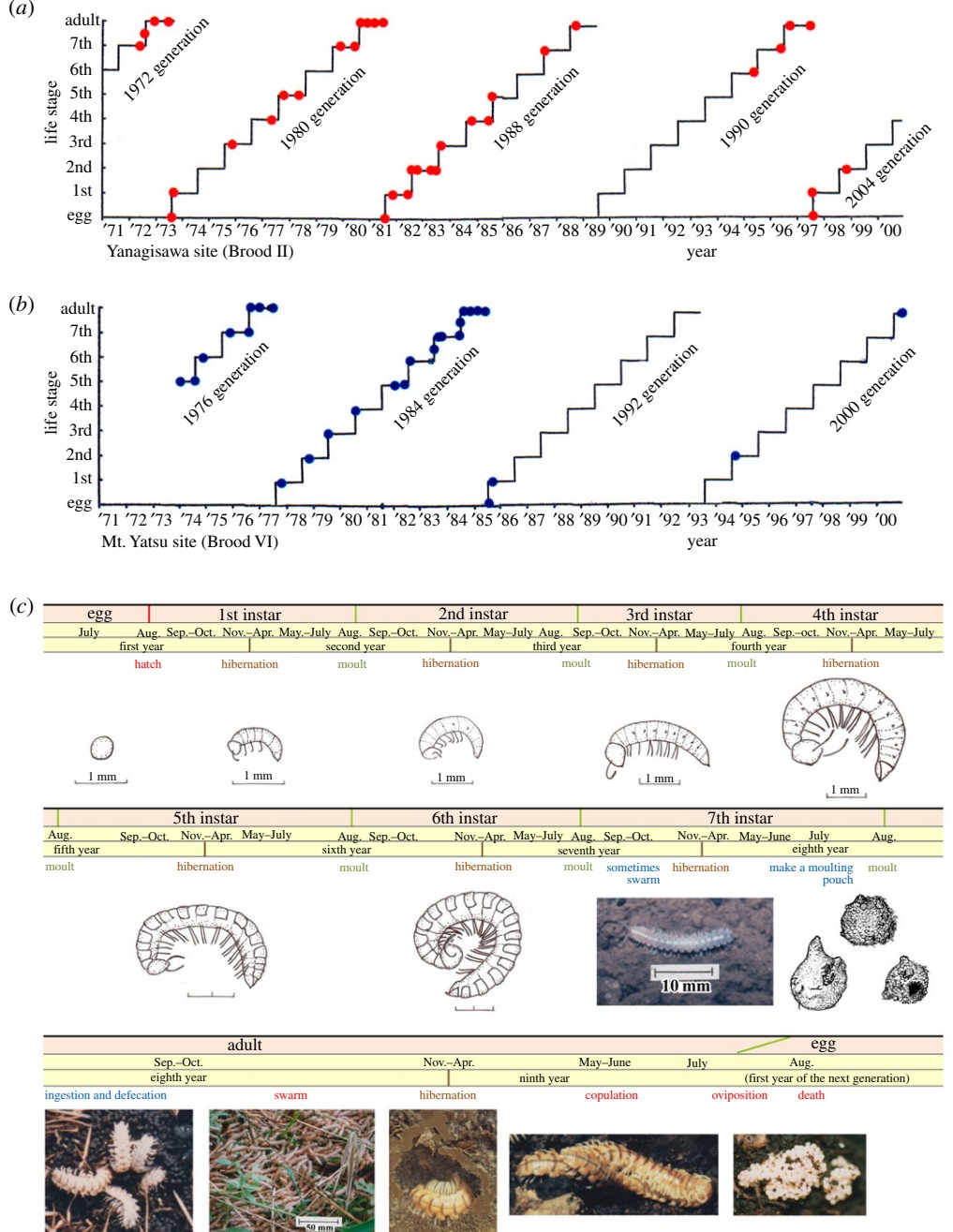

**Figure 5.** Life history of the train millipede (electronic supplementary material, tables S6 and S7). (*a,b*) Life stage of the millipede at each sampling time (red and blue dots) at the Yanagisawa site (*a*) and at the Mt. Yatsu site (*b*), respectively. (*c*) Schematic diagram of the 8-year life cycle. The millipedes moult once a year, need 7 years to transition from egg to adult and need one more year for maturation to lay eggs. Nymphs stay at a depth of 5–20 cm. The 7th instar occasionally swarm. Adults swarm in autumn, ingest litter and humus, hibernate in soil in winter. They copulate in the next spring, lay eggs in early summer and die in August. All illustrations were drawn by K. Niijima.

# 3. Results

## 3.1. Historical train obstructions and outbreaks

The oldest records of *P. l. a.* obstructions were in 1920 around the Niekawa Station on the Chuo Line (inaugurated in 1911) and in 1936 around the Kiyosato Station on the Koumi Line (inaugurated in 1935). Since then, many cases of train obstructions have been reported at intervals of 8 and/or 16 years (figure 4*a* and electronic supplementary material, table S5). During the mass emergence in the autumn of

1976, 1984 and 2000, aggregation was observed in a nearby road gutter (figure 4*b*), numerous dead millipedes were found on a nearby road (figure 4*c*) and extreme aggregation was also observed in a nearby deciduous broad-leaved forest (figure 4*d*). The train obstructions were then reported by several newspapers in Japan (e.g. the local Shinano-Mainichi Newspaper).

During the mass emergence of 1984 on the Koumi Line, the recorded density of train millipedes was 168–454 individuals per m$^2$ during 27 August 1984 and 11 July 1985 in nearby forests (point m2 in figure 2*a* and electronic supplementary material, table S6); in addition, the density of train millipedes in nearby roads was 768 individuals per m$^2$ on 27 August 1984 (point y1 in figure 2*a*) and 16–115 individuals per m$^2$ on 24–25 September 1984 (point m1 in figure 2*a*). Since the train obstruction of 1984 on the Koumi Line, mass emergence (swarming) has been reported every 8 years, but the last train obstruction was in 1984 (figure 4*a* and electronic supplementary material, tables S5 and S8.1).

At the Yanagisawa site (site 6 in figure 1*a* and electronic supplementary material, table S7), the highest density was recorded in the 1980 outbreak (the 7th-instar density in June 1980: 547 individuals per m$^2$; adult density in August 1980: 267 individuals per m$^2$; electronic supplementary material, table S7); swarming was observed in many locations in the forest in the autumn of 1980 (electronic supplementary material, table S8.2). Due to this extreme mass emergence, the construction of forest roads was cancelled for 3 days.

## 3.2. Life history of the train millipede

The train millipedes undertake a moulting in the summer every year and have seven larval instars. They become adults by the seven moulting after 7 years from egg deposition (figure 5 and electronic supplementary material, tables S6 and S7). The 1st instars appear in early August and move down 5–20 cm deep, where they hibernate and moult in the following summer. The 2nd to 7th instars also stay in soil during all seasons with the only following exception (electronic supplementary material, tables S6 and S7). Millipedes of the 7th instar (last before adult), may swarm on the ground on rare occasions (figure 5*c*).

The newly moulted adults emerge from their moulting pouches in late August. During September and October, the adults swarm on the soil surface at night and stay in soil during the daytime; though they also swarm in daytime on cloudy or rainy days in autumn (figures 4*b–d* and 5*c* and electronic supplementary material, tables S6 and S7). Because swarming distances were easily over 50 m, we sometimes did not find a single adult at the sampling point where hundreds of adults were seen several days before. However, in such a case, we found hundreds of adults aggregating along roadsides, gutters or nearby swamps (figure 4*b–d*). They copulate on or in the litter horizon in October. The millipedes then go down to a soil depth of 5–25 cm in December, coil their bodies and hibernate. In late spring, they become active and some creep out on the forest floor and copulate again. In July, they lay approximately 400–1000 eggs (200–300 egg mass two to three times; body dissection showed approx. 1000 eggs in the abdomen) and die within a month. In summary, this millipede needs 7 years from egg to adult and one more year for maturation. Thus, the 8-year periodicity of *P. l. a.* was confirmed by tracing the complete life history from eggs to adults in two different locations (the Mt. Yatsu and Yanagisawa sites) (figure 5 and electronic supplementary material, tables S6 and S7).

## 3.3. Categorization of the records into generations and broods

The life history of *P. l. a.* is summarized as follows. Adult females lay eggs on the border of litter and soil layers in summer (June–July). The eggs hatch a month later in July–August. Every year juveniles (nymphs) moult in summer reaching the adult stage in the seventh year. Juveniles stay in the same soil until 7th instar and adult stage (8th instar). Adults (rarely 7th instar) swarms in autumn, sometimes causing outbreaks. Adults copulate in autumn and next spring before/after hibernation. Adult females lay eggs in early summer. The identification of this species is only possible in adult males by checking male genitalia [10].

Suppose mass emergence (including outbreaks and swarming) was observed in the autumn of year $X$ and the following spring. Call it generation $X$. Then, we have '$X \pm 8n$' generations and eight independent lineages possible. Following the terminology of periodical cicadas [4,5], we call each of these eight lineages, brood. Based on these records, we term Brood I for the generations with the emergence in the autumn of 1971 and thereafter: Broods I (1971), II (1972), III (1973), IV (1974), V (1975), VI (1976), VII (1977) and VIII (1978). Note that the Mt. Yatsu site belongs to Brood VI and the Yanagisawa site, Brood II.

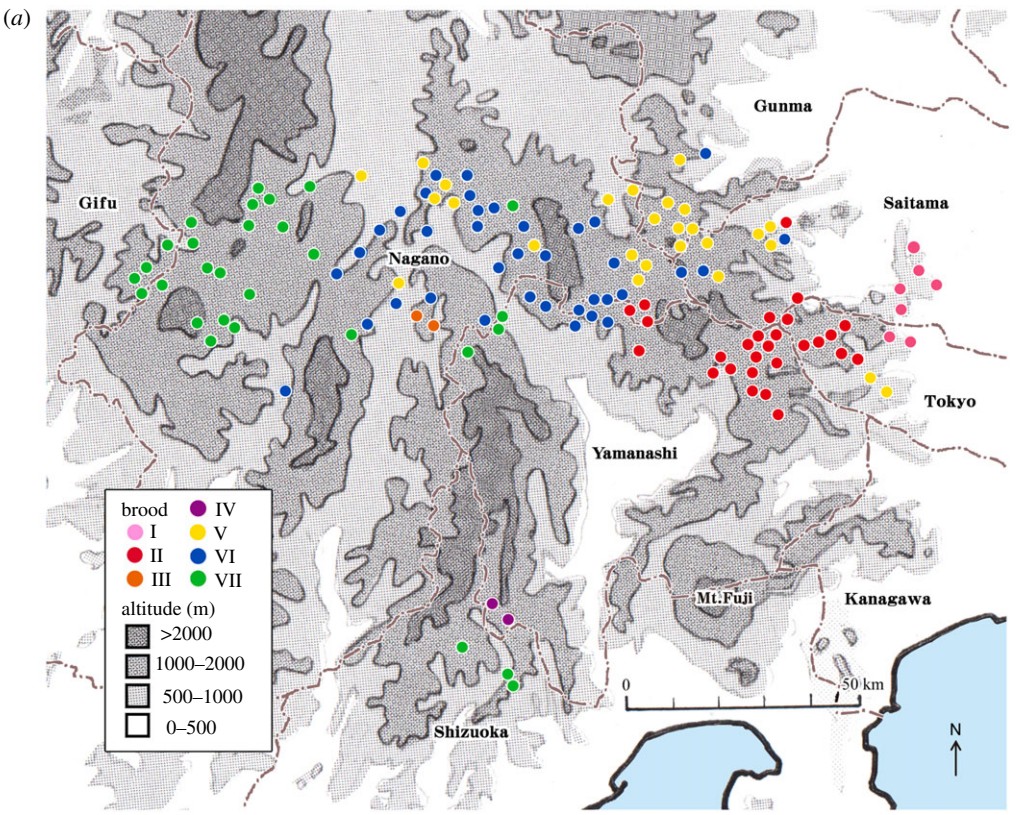

**Figure 6.** Distributions of *Parafontaria laminata armigera* and emergence records. (*a*) Distribution of broods. All broods were in the mountainous region of Central Japan. Brood VI (blue dots) was located in the centre of the brood distributions. Brood V (yellow) was located in the north, and Brood II (red) was in the southeast of Brood VI. Brood VII (green) distributed widely to the west and south of the brood distributions. Broods I (pink), III (orange) and IV (purple) were limited, and Brood VIII was not found. Elevation was traced from a 1/3 000 000 topographical map of 'Japan' published by the Geospatial Information Authority of Japan. (*b*) Confirmed generations (emergence year of adult). The coloured years show when the presence of adults *P. laminata armigera* was reported and grey years without any record on *P. l. a*. The generations of Brood VI were reported 12 since 1920; those of Broods V and II were seven since 1935 and 1956, respectively.

## 3.4. Distributions of broods

All seven broods, except Brood VIII, were distributed in the mountainous region of Central Japan (figure 6 and electronic supplementary material, table S8). Except for Brood I, which was located 500–1300 m.a.s.l., the others were found in 900–1800 m.a.s.l. There were a few reports of Brood VIII, but none of them could be confirmed.

Brood VI is located in the centre of the distribution (blue dots in figure 6*a* and electronic supplementary material, table S8.1). Almost all known train obstructions were caused by Brood VI. The railway track maintenance records during 1952–1992 confirm the 8-year periodical emergence (1952, 1960, 1968, 1976, 1984 and 1992) and no emergence between those years (figure 4). The nymphal data at both Mt. Yatsu

and Yanagisawa sites confirm the 8-year periodicity of *P. l. a.* (figure 5). Mass emergence or swarming of Brood VI was recorded for the emergence years of 1976, 1984, 1992, 2000, 2008 and 2016.

During 1912–1951, train obstructions were reported for the emergence years of Brood VI (1912, 1920, 1928 and 1936), but no record of emergence in 1944 because of the interruption by World War II (1941–1945). During 1912–2019, there were only two exceptions in the emergence of Brood VI. In 1943, train obstructions were reported between Kaioizumi and Nobeyama stations and between Shinanokawakami and Sakuuminokuchi stations in the Koumi Line. The former outbreak is suspected to be pre-adult (7th instar) outbreak and the latter, Brood V because it locates in the boundary of Broods V and VI. Thus, except two 1943 outbreaks, all positive (emergence) records and many negative records during 1912–2019 indicate the 8-year periodicity of Brood VI.

Brood II is distributed 600–1800 m.a.s.l., over almost the entire area of Chichibu-Tama-Kai National Park (including the Yanagisawa site) (red dots in figure 6*a* and point 6 in figure 1*a*). The millipedes live mainly in deciduous broad-leaved forests (electronic supplementary material, tables S8.2 and S8.6). Almost all generations since 1956 have been recorded. The first record of swarming of Brood II came from Mt. Kumotori (2017 m.a.s.l.). Swarming was observed not only at the peak of the mountain but also across the entire paths from the peak to Sanjonoyu Spa and to Aoiwa Cave (both approx. 1200 m.a.s.l.) on 3 October 1956 (The author reported the year incorrectly as 1966. We confirmed this error by contacting the author directly.) [20]. Our survey of nymphal development at the Yanagisawa site during 1972–1988 also agrees that there is 8-year periodical emergence (1972, 1980 and 1988) and nymphal stages from 1973 to 1987 except emergence years and a few missing years (due to the difficulty of finding nymphs) (figure 5*a* and electronic supplementary material, table S7).

The 8-year periodicity of Broods VI and II has been confirmed with all emergence years and numerous negative records, together with trace records of nymphal development. In the rest of the broods, we could only gather the emergence records.

Brood V is mainly located north of Brood VI (emerged 1 year earlier than Brood VI; yellow dots in figure 6*a*). The first record of Brood V was collected in 1935 at Suwa, Nagano Pref. (electronic supplementary material, table S8.3). This specimen became the type specimen (*Fontaia* (*Parafontaria* new subgenus) *armigera* n. sp.) for the train millipede (*Parafontaria laminata armigera*) [21]. However, the exact location of the collection within Suwa (very large area) was not recorded. The only train obstruction reported in Brood V was an obstruction in 1943 between the Sinano-kawakami and Saku-uminokuchi Stations, but it was accompanied with no samples or photos (electronic supplementary material, table S5).

Brood VII is mostly distributed in the west of Brood VI (green dots in figure 6 and electronic supplementary material, table S8.4). Some observations of swarming were reported in 1977 on Mt. Norikura [15] and mass emergence (no specimens and no photos) was reported from Kamitakara, Nyukawa and Takane Villages, Gifu Pref. (now all belonging to Takayama City) [22]. In 1985, many specimens of Brood VII were collected in Takane Village. The distribution of Broods V and VII also partially overlapped with that of Brood VI in the contact zone, since the simultaneous mass emergence of Brood V or VI (adult) and Brood VI or VII (the 7th instar) was confirmed in some locations [15]. The habitats of these three broods were mostly conifer plantations consisting of *Larix kaempferi* or *Cryptomeria japonica* (electronic supplementary material, tables S8.1, S8.3, S8.4 and S8.6).

The remaining three broods, I, III and IV, had very limited distributions, and the presence of them had been confirmed since 1973 when we began to work on the train millipedes. Brood I was distributed in the easternmost area of all the broods (pink dots in figure 6*a*, point 7 in figure 1*a* and electronic supplementary material, table S8.5.1). This brood appears mostly in Saitama Prefectural Natural Parks (Ryogami, Buko, Okumusashi, etc.), in the south site, bordering Chichibu-Tama-Kai National Park, where Brood II is found. Brood III was first found in 1973 in the Terasawa Experiment Forest, Shinshu University [15] (orange dots in figure 6*a* and electronic supplementary material, table S8.5.2). Since then, an 8-year emergence pattern has been reported until 2013 [15,23]. Brood IV (purple dots in figure 6*a* and electronic supplementary material, table S8.5.3) was far south and next to the isolated Brood VII. The first record of Brood IV came from Mt. Hatanagi of the southern part of the Japanese South Alps, Shizuoka City on October 1982 [9] (purple dots in figure 6*a*). On October 1990, over 50 specimens were collected at the Ikawa Experiment Station of Tsukuba University, Shizuoka City (by their staff), approximately 6 km east of the first location (electronic supplementary material, table S8.5.3).

## 4. Discussion

We confirmed the 8-year periodicity in the train millipede from the growth schedules at two field sites (figure 5 and electronic supplementary material, tables S6 and S7). The two reports of staff members in

the Japan National Railway (JNR) internal documents during the World War II apparently do not fit the 8-year periodicity (figure 4 and electronic supplementary material, table S5). If these reports during World War II are correct, these may be explained as follows: (i) one report of Brood VI may instead refer to the adjacent Brood V, which was not known in those days, and (ii) the other report of train obstruction in 1943 is suspected to be a large swarming of 7th instar nymphs that should have appeared in 1944 (figure 4 and electronic supplementary material, table S5). Unfortunately, we do not have any record of 1944 because of the disorder of the defeat in World War II in August 1945 (figure 4). Other than these two JNR reports, all reports and news records fell into the exact 8-year periodicity of the current distributions of all broods. Thus, the 8-year periodicity is widely confirmed in all broods (except these two unconfirmed cases), including all remaining historical records and recent survey records (figure 6 and electronic supplementary material, table S8).

This millipede is unique in that its moulting and oviposition are conditioned by winter low temperatures. Fujiyama showed that moulting requires the conditioning of winter low temperatures (approx. 80 days of 5°C for the 4th and 7th instars) [24]. He also showed that oviposition is initiated by 120 days but not by 60 days of 5°C [24], indicating that more than 60 days of cold temperatures are required for the initiation of egg laying. These results imply that both moulting and oviposition are conditioned by winter low temperatures, resulting in the yearly progress of instars and oviposition and consequently the 8-year life cycle. Note that winter soil temperatures in the studied millipede distribution were below 5°C at a depth of 10 cm for approximately five months (electronic supplementary material, table S4). This yearly progress is unique to *P. l. a.* (the train millipede), since many known millipedes moult twice or more per year [8,25,26]. For example, the life cycle of a montane millipede in the Central Alps, *Ochogona caroli*, which is distributed from 670 to 2000 m.a.s.l., is 2 years at 670 m and 3 years at 2000 m [27]. Therefore, as a periodical arthropod, the periodical train millipede exhibits an unusually long 8-year life cycle, very similar to periodical cicadas with 17- or 13-year life cycles [28] and comparable to many non-periodical temperate cicadas with life cycles less than 10 years [29].

*Parafontaria laminata armigera* contains defensive cyanogen in its body and releases cyanide when attacked [30]. Therefore, mass emergence in the millipede does not function in predation satiation or saturation, as in periodical cicadas.

The closely related *Parafontaria* species surround the current periodical millipede: *P. echizenensis* is west of Brood VII (point 8 in figure 1a), *P. kuhlgatzi* is north of Broods V and VI (points 9–12) and *P. laminata laminata* is on Mt. Fuji and on Boso peninsula (points 13, 14), south of Brood II. Those three species are distributed lower than 500 m in altitude: 300–880 m.a.s.l. (*P. e.*), 100–1900 m.a.s.l. (*P. k.*) and 200–1400 m.a.s.l. (*P. l. l.*) [9].

The central mountainous region of Japan (figures 1a and 6a), where *P. l. a.* is distributed in the elevations of 900–1800 m.a.s.l., is geologically the most active region with many large-scale crustal movements (the formation of Fossa Magna (dividing lines of eastern and western Japan), numerous ground upheavals due to volcano eruptions, etc.) [31]. These crustal movements took place 30–10 thousand years ago when the Earth had started cooling at the end of the interglacial period in the early Pliocene Epoch [32]. The train millipede, *P. l. a*, of which moulting requires conditioning by cooling, should have evolved though the double cooling effects of the global cooling (the glacial period) and local land cooling due to drastic ground upheavals in the areas. The evolution of the *Parafontaria* species group in relation to the geological history of Central Japan is a future topic of our interests, together with not only the phylogenetic relationship among this species group but also their life-history differences, e.g. life cycle lengths of these species, moulting and oviposition.

As a periodical animal, *P. l. a*, similar to periodical cicadas, demonstrates outbreaks or extreme mass emergence. This fact implies that the current train millipede has an elongated life history due to geological cooling during glacial periods, as in periodical cicadas [33]. However, the underlying mechanisms and evolutionary outcomes are different between the two cases. In periodical cicadas, the elongation of emergence cycles led to synchronous emergence (periodicity), which resulted in the selection of prime-numbered cycles (17- and 13-years) [34,35]. In the train millipede, the elongation of life cycles is suspected to be synchronized with winter hibernation. We also observe the separation of broods as in the periodical cicadas (figure 6) [4,5]. This mutual exclusiveness in the distributions of broods in both periodical cicadas and the train millipede is another future problem.

We have shown the existence of a periodical millipede, a new addition to periodical organisms with long life cycles: periodical cicadas, bamboos and some plants in the genus *Strobilanthes*. Even though swarming aggregations are very common in many millipedes, e.g. a common millipede *Pleuroloma flavipes* in the United States [8,25,36], *Parafontaria laminata armigera* is the first record of periodical non-insect arthropod.

Data accessibility. The datasets supporting this article are included in the main article and the electronic supplementary material.

Authors' contributions. K.N. collected filed data, media reports and performed experiments; J.Y. and K.N. analysed the data; M.N. and J.Y. constructed the manuscript and the figures; and all authors wrote the manuscript.

Competing interests. We declare we have no competing interests.

Funding. We received no funding for this study.

Acknowledgements. We thank the Forestry and Forest Products Research Institute (where K.N. belonged) and their staff members for supporting this project. We wish to thank the Water Resources Management Office, Bureau of Waterworks, Tokyo Metropolitan Government for the permission to study in the Forest for Water Conservation and for providing meteorological data at the Ochiai branch near the study sites. We also wish to thank the Nobeyama Office of Tsukuba University for the permission to study in the Experiment Forest of Tsukuba University and for providing meteorological data in the office. Special thanks are due to the late K. Shinohara for the identification of millipedes and guidance in the early studies. We thank M. Iwanami of the Tokyo Metropolitan Agriculture and Forest Research Center, for his assistance in the field survey. We also thank M. Kaijo and Y. Kuroda, both at Tsukuba University, M. Katakura and M. Okada, both at the Nagano Prefecture Forest Research Center, and K. Ishii, Y. Kuwabara, M. Terada, K. Yamamoto, T. Nohira and many other people for providing us with a great deal of information on the train millipede. We thank the late Seiji Uchiyama of the Japanese National Railway for detailed information about train obstructions by the millipede. We wish to thank Dr. Takuya Okabe for commenting on the entire manuscript.

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
