## [Reviewer comments · Royal Society Open Science]

Review History

RSOS-200198.R0 (Original submission)

Review form: Reviewer 1

Is the manuscript scientifically sound in its present form?

No

Are the interpretations and conclusions justified by the results?

No

Is the language acceptable?

Yes

Do you have any ethical concerns with this paper?

No

Have you any concerns about statistical analyses in this paper?

No

Recommendation?

Major revision is needed (please make suggestions in comments)

Comments to the Author(s)

I can't provide detailed comments on the methodology (I am not a field biologist), but it does seem like much more information should be provided in the main text. It's not clear to me how the sites were surveyed. From what is written, it's difficult to tell things such as how the authors are sure the surveys are all from the same brood, etc.

I provide more general feedback on the findings and conclusions below.

- Line 65: why does the need for a long period of low temperatures suggest an 8-year periodicity in particular? Doesn't this just suggest there's a relatively long life span?

- Related to the previous comment, this is again included in the Discussion (line 229). It's not clear why the conditions for molting add up to an 8 year cycle.

- I'm not entirely sure of the relevance of including the section "Historical train obstructions and outbreaks," since the purpose of the paper is to report that these millipedes are periodical. In general, some parts of the manuscript can be much more concise.

- Line 172: what are the F and HA horizons? Maybe I missed this.

- The second paragraph could be greatly expanded to provide more hypotheses (and ideas for future work) on why the millipedes have an 8-year cycle. Also, based on the map, it looks as though broods are relatively spatially segregated - as is the case for North American periodical cicadas. Do you have ideas on why this may be the case (also, is it possible that this is a consequence of how you collected data)?

- Relatedly, it's not always clear how much of a new discovery this paper adds in its current form. From reading this, it seems as though the 8-year periodicity has been suspected for some time. Perhaps elaborating on your methodology would help?

- In general, there's also a lot of room to expand on what might drive the behavior you observed. For example, there has been quite a bit of mathematical modeling work done in the case of periodical cicadas. Might that be useful here to explore hypotheses surrounding the 8-year cycles?

- The last paragraph of the discussion does not seem relevant to the rest of the paper. Also, the paper makes it sound like these millipedes - if anything - would be considered a pest species and not worth trying to conserve (though this may be totally inaccurate - it's just the feeling I had from the paper).

Review form: Reviewer 2

Is the manuscript scientifically sound in its present form?

No

Are the interpretations and conclusions justified by the results?

No

Is the language acceptable?

Yes

Do you have any ethical concerns with this paper?

No

Have you any concerns about statistical analyses in this paper?

No

Recommendation?

Major revision is needed (please make suggestions in comments)

Comments to the Author(s)

The paper hypothesizes that millipedes of the genus *Parafontaria* exhibit 8-year periodicity. This hypothesis, if true, would be notable because the phenomenon is rare in arthropods. However, the manuscript presents an unconvincing test of the hypothesis and presents anecdotal observations. A convincing test of their periodical nature would necessitate some rigorous data analyses and statistical test, which have not been provided. For example, a histogram of frequency of individuals occurring every eight years. Alternatively, especially if disparate sources are combined (including the historical newspaper data), a principal component analysis or similar might convey the results better. While the observations are interesting, and it does seem to be a periodical arthropod, the study lacks a synthesis, is much too long and detailed with disparate data sources, and reads instead as a rather disjointed collection of natural history observations and random environmental measurements.

The English grammar is somewhat problematic and inclusion of a native English speaking author who is a scientist to help tighten (significantly) and synthesize the disparate elements would be beneficial.

Miscellaneous comments:

Other xystodesmids "swarm", including *Pleurolooma flavipes* in the US. Mentioning this species would help give context to the present study.

It is unclear if adult millipedes are seen each year, which would not be consistent with a periodical nature.

It is unclear what the red dots represent in figure 2a and blue dots represent in 2b.

It is unclear what species Table 1 corresponds to.

Do not start sentences with abbreviated species name (P. l. a.).

Change fall 1984 to Fall 1984.

Decision letter (RSOS-200198.R0)

31-Mar-2020

Dear Professor Yoshimura:

Manuscript ID RSOS-200198 entitled "Eight-year periodical outbreaks of the train millipede" which you submitted to Royal Society Open Science, has been reviewed. The comments from reviewers are included at the bottom of this letter.

In view of the criticisms of the reviewers, the manuscript has been rejected in its current form. However, a new manuscript may be submitted which takes into consideration these comments.

Please note that resubmitting your manuscript does not guarantee eventual acceptance, and that your resubmission will be subject to peer review before a decision is made.

Your resubmitted manuscript should be submitted by 28-Sep-2020. If you are unable to submit by this date please contact the Editorial Office.

on behalf of Dr Jake Socha (Associate Editor) and Kevin Padian (Subject Editor)
openscience@royalsociety.org

Associate Editor Comments to Author (Dr Jake Socha):

The potential finding an 8-year periodical millipede is very interesting, but the current manuscript does not provide the quantitative data or analysis that is needed to properly assess the claim. In addition, there are issues with the manuscript in terms of its style, including being overly long, lacking a synthesis, and presenting as a collection of natural history notes. For these reasons, we are not able to consider the manuscript for further review.

Reviewers' Comments to Author:

Reviewer: 1

Comments to the Author(s)

I can't provide detailed comments on the methodology (I am not a field biologist), but it does seem like much more information should be provided in the main text. It's not clear to me how the sites were surveyed. From what is written, it's difficult to tell things such as how the authors are sure the surveys are all from the same brood, etc.

I provide more general feedback on the findings and conclusions below.

- Line 65: why does the need for a long period of low temperatures suggest an 8-year periodicity in particular? Doesn't this just suggest there's a relatively long life span?

- Related to the previous comment, this is again included in the Discussion (line 229). It's not clear why the conditions for molting add up to an 8 year cycle.

- I'm not entirely sure of the relevance of including the section "Historical train obstructions and outbreaks," since the purpose of the paper is to report that these millipedes are periodical. In general, some parts of the manuscript can be much more concise.

- Line 172: what are the F and HA horizons? Maybe I missed this.

- The second paragraph could be greatly expanded to provide more hypotheses (and ideas for future work) on why the millipedes have an 8-year cycle. Also, based on the map, it looks as though broods are relatively spatially segregated - as is the case for North American periodical cicadas. Do you have ideas on why this may be the case (also, is it possible that this is a consequence of how you collected data)?

- Relatedly, it's not always clear how much of a new discovery this paper adds in its current form. From reading this, it seems as though the 8-year periodicity has been suspected for some time. Perhaps elaborating on your methodology would help?

- In general, there's also a lot of room to expand on what might drive the behavior you observed. For example, there has been quite a bit of mathematical modeling work done in the case of periodical cicadas. Might that be useful here to explore hypotheses surrounding the 8-year cycles?

- The last paragraph of the discussion does not seem relevant to the rest of the paper. Also, the paper makes it sound like these millipedes - if anything - would be considered a pest species and not worth trying to conserve (though this may be totally inaccurate - it's just the feeling I had from the paper).

Reviewer: 2

Comments to the Author(s)

The paper hypothesizes that millipedes of the genus *Parafontaria* exhibit 8-year periodicity. This hypothesis, if true, would be notable because the phenomenon is rare in arthropods. However, the manuscript presents an unconvincing test of the hypothesis and presents anecdotal observations. A convincing test of their periodical nature would necessitate some rigorous data analyses and statistical test, which have not been provided. For example, a histogram of frequency of individuals occurring every eight years. Alternatively, especially if disparate sources are combined (including the historical newspaper data), a principal component analysis or similar might convey the results better. While the observations are interesting, and it does seem to be a periodical arthropod, the study lacks a synthesis, is much too long and detailed with disparate data sources, and reads instead as a rather disjointed collection of natural history observations and random environmental measurements.

The English grammar is somewhat problematic and inclusion of a native English speaking author who is a scientist to help tighten (significantly) and synthesize the disparate elements would be beneficial.

Miscellaneous comments:

Other xystodesmids "swarm", including *Pleuroloma flavipes* in the US. Mentioning this species would help give context to the present study.

It is unclear if adult millipedes are seen each year, which would not be consistent with a periodical nature.

It is unclear what the red dots represent in figure 2a and blue dots represent in 2b.

It is unclear what species Table 1 corresponds to.

Do not start sentences with abbreviated species name (P. l. a.).

Change fall 1984 to Fall 1984.

Author's Response to Decision Letter for (RSOS-200198.R0)

See Appendix A.

RSOS-201399.R0

Review form: Reviewer 3

Is the manuscript scientifically sound in its present form?

Yes

Are the interpretations and conclusions justified by the results?

Yes

Is the language acceptable?

Yes

Do you have any ethical concerns with this paper?

No

Have you any concerns about statistical analyses in this paper?

No

Recommendation?

Accept with minor revision (please list in comments)

Comments to the Author(s)

In this paper Niiijima et al. report the periodical emergence at 8-year intervals in populations of the millipede *Parafontaria laminata armigeres* in the mountain area of central Japan. Although the eight-year periodicity of this millipede has been reported in previous papers (Yoshida et al. 1985, 1987), Niiijima et al. provide comprehensive data for the life cycle and emergence pattern of this species based on long-term observation of populations and historical records of outbreaks,

which caused train obstruction, and confirms the 8 year periodicity. In animals, the periodical cicadas (*Magicicada* spp.) have been the almost only example of periodical species with life cycles longer than two years. Thus, *P. laminate armigeres* is the second example of periodical animals, providing a new material for the evolutionary study of unusual periodical cycle of organisms. While I have not found any major flaw in this study, the manuscript as it stands is difficult to follow, partly due to many miscellaneous information and data mentioned or presented in the main text. I recommend to move tables 1-3 to supplementary materials and revise the writing of the manuscript. Below I list places where changes may be needed.

L53-54: Please provide references for the statement: "there are many millipede species that adults survive several years in their life cycles".

L59: change to "Later, the genus *Fontaria* was renamed to *Parafontaria*, and the train millipedes were divided into..."

L80-82: The two sentences with "negative data" requires rewording. It is not logical that "These negative data include the historical outbreaks and train obstructions".

L91: during 1974 and 1985 (12 years consecutively) => during 1974-1985 (12 consecutive years)

L93-94: during 1972 and 1988 (17 years consecutively) => during 1972-1988 (17 consecutive years); the usage "during xxxx and xxxx" should be changed to "during xxxx-xxxx" in other places, too.

L94: (4 years cons.) => (4 consecutive years)

L95: change to "The detailed explanation"

L188: many reports => many cases

L189: in Fall => in the fall of

L190: near by => nearby

L205: in Fall 1980 => in the fall of 1980

L209: Revise this sentence. For example; The train millipedes undertake a moulting in the summer every year and have seven larval instars. They become adults by the eighth moulting after 8 years from egg deposition.

L212: "The seventh instars, which are the last instars," => "Millipedes of the seventh (last) instar"

L275: Change "Tanemura" and "him" to "the author".

L350-351: "exhibiting cooling adaptation in the molting trigger". I don't understand here. Is it "of which molting requires conditioning by cooling"?

Review form: Reviewer 4

Is the manuscript scientifically sound in its present form?

Yes

Are the interpretations and conclusions justified by the results?

Yes

Is the language acceptable?

Yes

Do you have any ethical concerns with this paper?

No

Have you any concerns about statistical analyses in this paper?

No

Recommendation?

Accept with minor revision (please list in comments)

Comments to the Author(s)

This manuscript presents long-term data from two study sites, as well as historical data from a wider area, to confirm the hypothesis that train millipedes (*Parafontaria laminata armigera*) follow an 8-year periodical life cycle. I found the work convincing and extremely interesting. Moreover, I hope documenting this phenomenon will be an important first step towards understanding its eco-evolutionary underpinnings, which (in my view) would be even more interesting.

Minor suggestions.

Line 54. replace "that" by "in which"

Line 259. The sentence that begins with "Our survey of..." is not grammatical and possibly incomplete.

Line 304. replace "blood" by "brood"

Line 305 and elsewhere. Replace "plot" by "dot" throughout.

Figure 1 and Line 483. Since other species don't seem strictly relevant to the topic at hand, I was briefly wondering if it may be better to remove such information. However I don't insist on this, as I can imagine circumstances where this info could be useful for other researchers.

Line 528. Delete "are"

Line 538. At the beginning of the line, replace "brood" by "broods".

Line 543. Tables 1-3 may be of little interest to the average reader, so perhaps could be moved to the supplementary material.

Decision letter (RSOS-201399.R0)

Dear Professor Yoshimura

The Editors assigned to your paper RSOS-201399 "Eight-year periodical outbreaks of the train millipede" have now received comments from reviewers and would like you to revise the paper in accordance with the reviewer comments and any comments from the Editors. Please note this decision does not guarantee eventual acceptance.

Please submit your revised manuscript and required files (see below) no later than 21 days from today's (ie 19-Oct-2020) date. Note: the ScholarOne system will 'lock' if submission of the revision

is attempted 21 or more days after the deadline. If you do not think you will be able to meet this deadline please contact the editorial office immediately.

on behalf of Dr Jake Socha (Associate Editor) and Kevin Padian (Subject Editor)
openscience@royalsociety.org

Associate Editor Comments to Author (Dr Jake Socha):

Both reviewers have positive views of the manuscript, but there are still a few things to address. In your revision, please in particular pay attention to reviewer 3's comment that the manuscript was difficult to follow, and make changes accordingly.

Reviewer comments to Author:

Reviewer: 3
Comments to the Author(s)

In this paper Niiijima et al. report the periodical emergence at 8-year intervals in populations of the millipede *Parafontaria laminate armigeres* in the mountain area of central Japan. Although the eight-year periodicity of this millipede has been reported in previous papers (Yoshida et al. 1985, 1987), Niiijima et al. provide comprehensive data for the life cycle and emergence pattern of this species based on long-term observation of populations and historical records of outbreaks, which caused train obstruction, and confirms the 8 year periodicity. In animals, the periodical cicadas (*Magicicada* spp.) have been the almost only example of periodical species with life cycles longer than two years. Thus, *P. laminate armigeres* is the second example of periodical animals, providing a new material for the evolutionary study of unusual periodical cycle of organisms. While I have not found any major flaw in this study, the manuscript as it stands is difficult to follow, partly due to many miscellaneous information and data mentioned or presented in the main text. I recommend to move tables 1-3 to supplementary materials and revise the writing of the manuscript. Below I list places where changes may be needed.

L53-54: Please provide references for the statement: "there are many millipede species that adults survive several years in their life cycles".

L59: change to "Later, the genus *Fontaria* was renamed to *Parafontaria*, and the train millipedes were divided into..."

L80-82: The two sentences with "negative data" requires rewording. It is not logical that "These negative data include the historical outbreaks and train obstructions".

L91: during 1974 and 1985 (12 years consecutively) => during 1974-1985 (12 consecutive years)
 L93-94: during 1972 and 1988 (17 years consecutively) => during 1972-1988 (17 consecutive years); the usage "during xxxx and xxxx" should be changed to "during xxxx-xxxx" in other places, too.

L94: (4 years cons.) => (4 consecutive years)

L95: change to "The detailed explanation"

L188: many reports => many cases

L189: in Fall => in the fall of

L190: near by => nearby

L205: in Fall 1980 => in the fall of 1980

L209: Revise this sentence. For example; The train millipedes undertake a moulting in the summer every year and have seven larval instars. They become adults by the eighth moulting after 8 years from egg deposition.

L212: "The seventh instars, which are the last instars," => "Millipedes of the seventh (last) instar"

L275: Change "Tanemura" and "him" to "the author".

L350-351: "exhibiting cooling adaptation in the molting trigger". I don't understand here. Is it "of which molting requires conditioning by cooling"?

Reviewer: 4

Comments to the Author(s)

This manuscript presents long-term data from two study sites, as well as historical data from a wider area, to confirm the hypothesis that train millipedes (*Parafontaria laminata armigera*) follow an 8-year periodical life cycle. I found the work convincing and extremely interesting. Moreover, I hope documenting this phenomenon will be an important first step towards understanding its eco-evolutionary underpinnings, which (in my view) would be even more interesting.

Minor suggestions.

Line 54. replace "that" by "in which"

Line 259. The sentence that begins with "Our survey of..." is not grammatical and possibly incomplete.

Line 304. replace "blood" by "brood"

Line 305 and elsewhere. Replace "plot" by "dot" throughout.

Figure 1 and Line 483. Since other species don't seem strictly relevant to the topic at hand, I was briefly wondering if it may be better to remove such information. However I don't insist on this, as I can imagine circumstances where this info could be useful for other researchers.

Line 528. Delete "are"

Line 538. At the beginning of the line, replace "brood" by "broods".

Line 543. Tables 1-3 may be of little interest to the average reader, so perhaps could be moved to the supplementary material.

===PREPARING YOUR MANUSCRIPT===

===PREPARING YOUR REVISION IN SCHOLARONE===

- Ensure that your data access statement meets the requirements at <https://royalsociety.org/journals/authors/author-guidelines/#data>. You should ensure that you cite the dataset in your reference list. If you have deposited data etc in the Dryad repository, please include both the 'For publication' link and 'For review' link at this stage.
- If you are requesting an article processing charge waiver, you must select the relevant waiver option (if requesting a discretionary waiver, the form should have been uploaded at Step 3 'File upload' above).
- If you have uploaded ESM files, please ensure you follow the guidance at <https://royalsociety.org/journals/authors/author-guidelines/#supplementary-material> to include a suitable title and informative caption. An example of appropriate titling and captioning may be found at https://figshare.com/articles/Table_S2_from_Is_there_a_trade-off_between_peak_performance_and_performance_breadth_across_temperatures_for_aerobic_sc_ope_in_teleost_fishes_/3843624.

Author's Response to Decision Letter for (RSOS-201399.R0)

See Appendix B.

RSOS-201399.R1 (Revision)

Review form: Reviewer 3

Is the manuscript scientifically sound in its present form?

Yes

Are the interpretations and conclusions justified by the results?

Yes

Is the language acceptable?

Yes

Do you have any ethical concerns with this paper?

No

Have you any concerns about statistical analyses in this paper?

No

Recommendation?

Accept as is

Comments to the Author(s)

The authors have made changes according to reviewers' suggestions. The revised manuscript would be acceptable.

Review form: Reviewer 4**Is the manuscript scientifically sound in its present form?**

Yes

Are the interpretations and conclusions justified by the results?

Yes

Is the language acceptable?

Yes

Do you have any ethical concerns with this paper?

No

Have you any concerns about statistical analyses in this paper?

No

Recommendation?

Accept with minor revision (please list in comments)

Comments to the Author(s)

I am satisfied with the revision, but have noticed some small linguistic errors in the revised text (see below).

L322: replace "apparently does fit" with "apparently do not fit"

L324 replace "is the mistake of...days" with "may instead refer to the adjacent brood V, which was not known in those days"

L325 replace "on 1943" with "in 1943"

L327 replace "of the defeated" with "of the defeat in"

L337 replace "Thus ... including" with "Thus, the 8-year periodicity is widely confirmed in all broods (except these two unconfirmed cases), including"

Decision letter (RSOS-201399.R1)

Dear Professor Yoshimura

On behalf of the Editors, we are pleased to inform you that your Manuscript RSOS-201399.R1 "Eight-year periodical outbreaks of the train millipede" has been accepted for publication in Royal Society Open Science subject to minor revision in accordance with the referees' reports. Please find the referees' comments along with any feedback from the Editors below my signature.

Please submit your revised manuscript and required files (see below) no later than 7 days from today's (ie 11-Dec-2020) date. Note: the ScholarOne system will 'lock' if submission of the revision is attempted 7 or more days after the deadline. If you do not think you will be able to meet this deadline please contact the editorial office immediately.

on behalf of Dr Jake Socha (Associate Editor) and Kevin Padian (Subject Editor)
openscience@royalsociety.org

Associate Editor Comments to Author (Dr Jake Socha):

The reviewers have no further major comments and are satisfied with the revisions. Congratulations on the acceptance of this manuscript. However, please consider addressing reviewer 1's few comments to address language issues prior to final publication.

Reviewer comments to Author:

Reviewer: 4
Comments to the Author(s)

I am satisfied with the revision, but have noticed some small linguistic errors in the revised text (see below).

L322: replace "apparently does fit" with "apparently do not fit"

L324 replace "is the mistake of...days" with "may instead refer to the adjacent brood V, which was not known in those days"

L325 replace "on 1943" with "in 1943"

L327 replace "of the defeated" with "of the defeat in"

L337 replace "Thus ... including" with "Thus, the 8-year periodicity is widely confirmed in all broods (except these two unconfirmed cases), including"

Reviewer: 3
 Comments to the Author(s)

The authors have made changes according to reviewers' suggestions. The revised manuscript would be acceptable.

===PREPARING YOUR MANUSCRIPT===

===PREPARING YOUR REVISION IN SCHOLARONE===

Author's Response to Decision Letter for (RSOS-201399.R1)

See Appendix C.

Decision letter (RSOS-201399.R2)

This year has been very difficult for everyone, and we want to take the opportunity to thank you for your continued support in 2020.

With our best for a peaceful festive period and New Year, and we look forward to working with you in 2021.

Dear Professor Yoshimura,

It is a pleasure to accept your manuscript entitled "Eight-year periodical outbreaks of the train millipede" in its current form for publication in Royal Society Open Science.

on behalf of Dr Jake Socha (Associate Editor) and Kevin Padian (Subject Editor)
openscience@royalsociety.org

Appendix A

Associate Editor Comments to Author (Dr Jake Socha):

The potential finding an 8-year periodical millipede is very interesting, but the current manuscript does not provide the quantitative data or analysis that is needed to properly assess the claim. In addition, there are issues with the manuscript in terms of its style, including being overly long, lacking a synthesis, and presenting as a collection of natural history notes. For these reasons, we are not able to consider the manuscript for further review.

REPLY: Tremendously thank you very much for your constructive comments. We are deeply grateful in these comments/suggestions. In your comments, we marked the points in yellow background we especially care for. We rewrote the structure of the manuscript entirely, so that the manuscript is readable and understandable.

For quantitative data and analysis, we include the followings:

- (1) We make sure to add/correct the manuscript to show the negative (observation) data in the non-period years is included. All emergence/non-emergence data except two cases(see below) agree the 8-year periodicity in the observational data. We explicitly state this fact of data. Note the two cases are (i) suspected miss-identification of broods at the boundary of two broods and (ii) suspected 7-instar swarming. These are explained in the text.
- (2) Life history tracing observations over 1 generation at both the Mt. Yatsu and the Yanagisawa sites (Figure 5) confirms 8-year life cycles in this millipede. Our tracking survey of nymphs during the development from adult-egg-juveniles-adult confirms 8-year life cycles for the two observed broods. The molting is also found to take place in the summer triggered by the winter cold. This indicates that this 8-year life cycle is fixed in this millipede species.

For the presentation of the collections of natural history, we restructured the entire manuscript and supplementary material, so that the resultant manuscript focus on the periodicity proof in the ordinary style of a research paper. We hope this radical change is acceptable to you.

Reviewers' Comments to Author:

Reviewer: 1

Comments to the Author(s)

I can't provide detailed comments on the methodology (I am not a field biologist), but it does seem like much more information should be provided in the main text.

It's not clear to me how the sites were surveyed. From what is written, it's difficult to tell things such as how the authors are sure the surveys are all from the same brood, etc.

REPLY: Thank you deeply for your constructive comments. In your comments, we marked **the points in yellow background** we especially care for. We rewrote the structure of the manuscript entirely, so that the manuscript is readable and understandable. We especially focus on the survey methods and data. We also included the negative data explicitly, so that the 8-year periodicity is evident.

I provide more general feedback on the findings and conclusions below.

- Line 65: why does the need for a long period of low temperatures suggest an 8-year periodicity in particular? Doesn't this just suggest there's a relatively long life span?

REPLY: Thank you. We rewrite to explain the reason in detail. See below.

- Related to the previous comment, this is again included in the Discussion (line 229). It's not clear why the conditions for molting add up to an 8 year cycle.

REPLY: Thank you for your comments. We rewrote this, so that the 8-year periodicity is clearly expected/implicated from Fujiyama's experimental work [24].

LINES 69-72: Secondly four-year consecutive survey indicated that *P. l. a.* molts once a year in summer (due to the need for long cold winter for molting) and becomes adults in the 8th year, because millipedes belonging to the order Polydesmida had been known to molt 7 times to be adult in the 8th instar [11,12].

LINES 320-323: Fujiyama showed that molting requires the conditioning of winter low temperatures (approximately 80 days of 5°C for the 4th and 7th instars) [24]. He also showed that oviposition is initiated by 120 days but not by 60 days of 5°C [24], indicating more than 60 days of cold temperatures are required for the initiation of egg laying.

- I'm not entirely sure of the relevance of including the section "Historical train obstructions and outbreaks," since the purpose of the paper is to report that these millipedes are periodical. In general, some parts of the manuscript can be much more concise.

REPLY: Thank you for your comments. We remove the section and only use the part that is related to the 8-year outbreak records.

- Line 172: what are the F and HA horizons? Maybe I missed this.

REPLY: Thank you for your comments. This part of natural history description is entirely cut in the revision because it does not relate to the 8-year periodicity. Note that these are pedology terms: Fermented horizon (F) and Humus A horizon (HA) are lower litter layers and top soil layer of the soil layers (layer is called horizon in pedology).

- The second paragraph could be greatly expanded to provide more hypotheses (and ideas for future work) on why the millipedes have an 8-year cycle. Also, based on the map, it looks as though broods are relatively spatially segregated - as is the case for North American periodical cicadas. Do you have ideas on why this may be the case (also, is it possible that this is a consequence of how you collected data?)?

REPLY: Thank you for your comments. In this paper, we focus on the confirmation of the 8-year periodicity only. We here include the statement of negative data to confirm the 8-year periodicity. We also include the tracking of the life history from adult-egg-nymphs-adult in two locations (the Mt. Yatsu and Yanagisawa sites) in the revision. For the future work, we like to consider the evolution of 8-year periodicity from shorter life cycles of other *Parafontaria* species. We are preparing the next article in this topic including the evolutionary change in life cycle length (some of them are our originals). Thus, we decide to mention only the evolutionary history of this species group as a future topic in Lines 339-356.

For the separation of broods in this millipede is also the future topic [Lines 364-366]. Actually, the last author Jin Yoshimura is preparing a manuscript proposing the mechanisms of spatial brood exclusion now (hoping to publish within this year) and its mechanism is quite unique and not likely to be the same with the current 8-year millipede.

- Relatedly, it's not always clear how much of a new discovery this paper adds in its current form. From reading this, it seems as though the 8-year periodicity has been suspected for some time. Perhaps elaborating on your methodology would help?

REPLY: Thank you for your comments. We move the important method from Supplementary material to the main text. We hope it is now clear that our confirmation of the 8-year periodicity in this revision.

- In general, there's also a lot of room to expand on what might drive the behavior you observed. For example, there has been quite a bit of mathematical modeling work done in the case of periodical cicadas. Might that be useful here to explore hypotheses surrounding the 8-year cycles?

REPLY: Thank you for your comments. We like to divulge this topic in future when we study the *Parafontaria* species group.

- The last paragraph of the discussion does not seem relevant to the rest of the paper. Also, the paper makes it sound like these millipedes - if anything - would be considered a pest species and not worth trying to conserve (though this may be totally inaccurate - it's just the feeling I had from the paper).

REPLY: Thank you for your comments. In the revision we omitted the paragraph entirely because it is not relevant to the 8-year periodicity. This millipede and all the *Parafontaria* species group are the beneficial arthropods in forests processing the litter to decomposed soils.

Reviewer: 2

Comments to the Author(s)

The paper hypothesizes that millipedes of the genus *Parafontaria* exhibit 8-year periodicity. This hypothesis, if true, would be notable because the phenomenon is rare in arthropods. However, the manuscript presents an unconvincing test of the hypothesis and presents anecdotal observations. A convincing test of their periodical nature would necessitate some rigorous data analyses and statistical test, which have not been provided. For example, a histogram of frequency of individuals occurring every eight years. Alternatively, especially if disparate sources are combined (including the historical newspaper data), a principal component analysis or similar might convey the results better. While the observations are interesting, and it does seem to be a periodical arthropod, the study lacks a synthesis, is much too long and detailed with disparate data sources, and reads instead as a rather disjointed collection of natural history observations and random environmental measurements.

REPLY: Thank you for your great comments. We eliminate all other natural history components and focus on the proof of the 8-year periodicity only in *Parafontaria laminata armigera* (the true train millipede). In our unpublished data, other species in this genus have a shorter life cycle. We actually preparing the next paper about the evolution of the *Parafontaria* species group (2 species and 1 species with two

subspecies: in total of 4 taxa) in the next manuscript. As suggested by the handling editor, we extensively rewrote to focus on the proof/confirmation of the 8-year periodicity in *Parafontaria laminata armigera* (the true train millipede). Here we have two proofs: (i) complete tracking observations of life cycles (adult-egg-several instars nymphs-adult) in two different broods (locations: the Mt. Yatsu and Yanagisawa sites), and (ii) successive observations of emergence/non-emergence (combined positive/negative data) that demonstrate the 8-year periodicity (except two observations) . These two cases are suspected to be (1) different broods and (2) 7th instar swarming. If we exclude these two unsure cases, all combined positive/negative data shows no exception; we cannot carry any statistics. Note that if we observe any one evident outlier observation, we have to deny the 8-year periodicity in this millipede. We believe that the statistics is not useful in this periodicity confirmation.

We hope that our reconstruction of the whole structure of the manuscript now solves your comment about the presentation of the paper. Note we remove a lot of natural history notes not related to the 8-year periodicity.

The English grammar is somewhat problematic and inclusion of a native English speaking author who is a scientist to help tighten (significantly) and synthesize the disparate elements would be beneficial.

REPLY: Thank you for your comments. The manuscript had been already checked by the Springer Nature Editing Service. The reason why it is difficult to read is the way the manuscript is structured is not the usual method, but a way much rhetorical presentation. Here we remove the rhetorical structure and the unrelated natural history notes to make it the standard research article. We hope that this modification is enough to you.

Miscellaneous comments:

Other xystodesmids "swarm", including *Pleuroloma flavipes* in the US. Mentioning this species would help give context to the present study.

REPLY: Thank you for your comments. We include citation in the text:

Lines 369-371: Even though swarming aggregations are very common in many millipedes [12, 25, 36], *Parafontaria laminata armigera* is the first record of periodical non-insect arthropod.

It is unclear if adult millipedes are seen each year, which would not be consistent with a periodical nature.

REPLY: Thank you for your comments. In each location, this species (brood) is not found except brood emergence year (and the year before as 7th instar nymphs). We may find other species of millipedes in every year.

It is unclear what the red dots represent in figure 2a and blue dots represent in 2b.

REPLY: Thank you for your comments. We add the explanation for the dots: Figure 5. Life history of the train millipede (tables 1, 2). (*a*, *b*) Life stage of the millipede at each sampling time (red and blue dots) at the Yanagisawa site (*a*) and at the Mt. Yatsu site (*b*), respectively.

It is unclear what species Table 1 corresponds to.

REPLY: Thank you for your comments. We are only dealing with *Parafontaria laminata armigera*, the true train millipede. In the revision, it is a part of Figure 6 and we state the species name in the figure legends.

Do not start sentences with abbreviated species name (P. l. a.).

REPLY: Thank you for your comments. We remove this abbreviation in the beginning of a sentence.

Change fall 1984 to Fall 1984.

REPLY: Thank you for your comments. Done.

Appendix B

One-to-one responses to the reviewers' comments

Associate Editor Comments to Author (Dr Jake Socha):

Both reviewers have positive views of the manuscript, but there are still a few things to address. In your revision, please in particular pay attention to reviewer 3's comment that the manuscript was difficult to follow, and make changes accordingly.

Reply-to-Editor: Thank you for your comments. We are very glad to see the positive views on the manuscript. We also agree that tables 1-3 should be in Supplementary Information.

Reviewer comments to Author:

Reviewer: 3

Comments to the Author(s)

In this paper Nijima et al. report the periodical emergence at 8-year intervals in populations of the millipede *Parafontaria laminata armigera* in the mountain area of central Japan. Although the eight-year periodicity of this millipede has been reported in previous papers (Yoshida et al. 1985, 1987), Nijima et al. provide comprehensive data for the life cycle and emergence pattern of this species based on long-term observation of populations and historical records of outbreaks, which caused train obstruction, and confirms the 8 year periodicity. In animals, the periodical cicadas (*Magicicada* spp.) have been the almost only example of periodical species with life cycles longer than two years. Thus, *P. laminata armigera* is the second example of periodical animals, providing a new material for the evolutionary study of unusual periodical cycle of organisms.

While I have not found any major flaw in this study, the manuscript as it stands is difficult to follow, partly due to many miscellaneous information and data mentioned or presented in the main text. I recommend to move tables 1-3 to supplementary materials and revise the writing of the manuscript. Below I list places where changes may be needed.

Reply 3-1. Thank you for your constructive comments. We follow all of your suggestions and revise the manuscript accordingly. We agree and are very happy to move tables 1-3 to supplementary materials. (tables 1-3 → table S6-S8) (table S2 →S4, table S3→S2, table S4→S3).

Corrected copy: P3 L95, 98, 102; P4 L109, 119, 126, 131, 140, 141; P6 L217, 220, 225, 235; P8 L254, 260, 281; P9 L289, 297, 303, 311, 316, 319, 320; P10 L325, 329, 349.

Clean copy: P3 L95, 98, 102; P4 L109, 119, 126, 131, 140, 141; P6 L217, 220, 225, 235; P8 L254, 260, 281; P9 L289, 297, 303, 311, 316, 319, 320; P10 L325, 329, 349.

L53-54: Please provide references for the statement: “there are many millipede species that adults survive several years in their life cycles”.

Reply 3-2. Blower, JG (1985) [ref. No. 8 in the revised text] is cited for this statement.

Corrected copy: P2 L54, 64-65, 69; P3 L74, 76; P5 L154, 159, 174; P6 L186; P7 L225; P9 L320; P10 L350; P11 L365, 393. Clean copy: P2 L54, 64, 68; P3 L73, 75; P5 L150, 155, 170; P6 L182; P7 L220; P9 L312; P10 L342; P11 L357, 284.

L59: change to “Later, the genus Fontaria was renamed to Parafontaria, and the train millipedes were divided into...”

Reply 3-3. Done. Corrected copy: P2 L60-61. Clean copy: P2 L60.

L80-82: The two sentences with “negative data” requires rewording. It is not logical that “These negative data include the historical outbreaks and train obstructions”.

Reply 3-4. Thanks a lot. Certainly, it was difficult to understand or logically wrong. We found two reports from JR managers does not fit to 8-year periodicity (looks like negative records). We found them are also explainable, if not wrong reports. Here we delete them and put them in discussion. We corrected the sentences as follows: “We also gather historical outbreaks and train obstructions of this millipede between 1920 and 2016.” Corrected copy: P3 L82-85.

Corrected copy: P3 L81-82.

In the second sentence of the discussion section: “The two reports of staff members in the Japan National Railway (JNR) internal documents during the World War II apparently does fit the 8-year periodicity (Fig. 4, table S5). If these reports during the WWII are correct, these may be explained as follows: (1) One report of brood VI is the mistake of brood V because the adjacent brood V was not known in those days; (2) the other report of train obstruction on 1943 is suspected to be a large swarming of 7th instar nymphs that should have appeared in 1944 (figure 4, table S5). Unfortunately, we do not have any record of 1944, because of the disorder of the defeated WWII in August 1944 (figure 4). Other than these two JNR reports, all reports and news records fell into the exact 8-year periodicity of the current distributions of all

broods. Thus, the conformation of reports (except these two unconfirmed cases) strongly indicates that the 8-year periodicity is widely confirmed in all broods including all remaining historical records and recent survey records (figure 6)." Corrected copy: P10 L329-339.

Clean copy: P9 L321-P10 331.

L91: during 1974 and 1985 (12 years consecutively) => during 1974-1985 (12 consecutive years)

Reply 3-5. Done in all applicable sentences. Corrected copy: P3 L94, 97. Clean copy: P3 L91, 94.

L93-94: during 1972 and 1988 (17 years consecutively) => during 1972-1988 (17 consecutive years); the usage "during xxxx and xxxx" should be changed to "during xxxx-xxxx" in other places, too.

Reply 3-6. Done in all applicable sentences. Corrected copy: P3 L94, 97; P8 L270, 272, 277.

Clean copy: P3 L91, 94; P8 L262, 264, 269.

L94: (4 years cons.) => (4 consecutive years)

Reply 3-7. Done. Corrected copy: P3 L98. Clean copy: P3 L94.

L95: change to "The detailed explanation"

Reply 3-8. Done. Corrected copy: P3 L99. Clean copy: P3 L95.

L188: many reports => many cases

Reply 3-9. Done. Corrected copy: P6 L193. Clean copy: P6 L189.

L189: in Fall => in the fall of

Reply 3-10. Done. Corrected copy: P4 L120, 121; P6 L194. Clean copy: P4 L116, 117; P6 L190.

L190: near by => nearby

Reply 3-11. Done. Corrected copy: P6 L195. Clean copy: P6 L191.

L205: in Fall 1980 => in the fall of 1980

Reply 3-12. Done. Corrected copy: P6 L210; P7 L249-250. Collected copy: P6 L206; P7 L244-245

L209: Revise this sentence. For example; The train millipedes undertake a moulting in the summer every year and have seven larval instars. They become adults by the eighth moulting after 8 years from egg deposition.

Reply 3-13. We have changed following your suggestions. Thank you indeed! We replace the sentence as suggested: The train millipedes undertake a molting in the summer every year and

have seven larval instars. They become adults by the eighth molting after 8 years from egg deposition (figure 5, tables S6, S7). Corrected copy: P6 L216-P7 L217. Clean copy: P6 L211.

L212: “The seventh instars, which are the last instars,” => “Millipedes of the seventh (last) instar”

Reply 3-14. Done. Corrected copy: P7 L220. Clean copy: P7 L215.

L275: Change “Tanemura” and “him” to “the author”.

Reply 3-15. Done. Corrected copy: P8 L285-286. Clean copy: P8 L276-277.

L350-351: “exhibiting cooling adaptation in the molting trigger”. I don’t understand here. Is it “of which molting requires conditioning by cooling”?

Reply 3-16. Thanks a lot. Your rewrite is correct. We have changed as suggested.

Corrected copy: P11 L372-373. Clean copy: P11 L364-365.

Reviewer: 4

Comments to the Author(s)

This manuscript presents long-term data from two study sites, as well as historical data from a wider area, to confirm the hypothesis that train millipedes (*Parafontaria laminata armigera*) follow an 8-year periodical life cycle. I found the work convincing and extremely interesting. Moreover, I hope documenting this phenomenon will be an important first step towards understanding its eco-evolutionary underpinnings, which (in my view) would be even more interesting.

Reply 4-1. Thank you for your comments. We revised the manuscript accordingly, except Figure 1, which we like to keep in the main text for Japanese millipede and soil biologist readers.

Minor suggestions.

Line 54. replace “that” by “in which”

Reply 4-2. Done. Corrected copy: P2 L54. Clean copy: P2 L54.

Line 259. The sentence that begins with “Our survey of...” is not grammatical and possibly incomplete.

Reply 4-3. Thanks a lot. This sentence is certainly mysterious. Our meaning is only the data at Mt Yatsu and Yanagisawa both confirms the 8-year periodicity of *Parafontaria laminata armigera*. We have changed as follows: The nymphal data at both Mt. Yatsu and Yanagisawa

sites confirm the 8-year periodicity of *P. l. a.* (figure 5). Corrected copy: P8 L266-267. Clean copy: P8 L258-259.

Line 304. replace “blood” by “brood”

Reply 4-4. Done. Corrected copy: P9 L314. Clean copy: P9 L306.

Line 305 and elsewhere. Replace “plot” by “dot” throughout.

Reply 4-5. Replace plot by dot in all sentences. Corrected copy: P8 L260, 280; P9 L295-296, 303,315, 319, 320, P16 L543. Clean copy: P8 L255, 272; P9 L287-288, 295,307, 311, 312, P20 L534.

Figure 1 and Line 483. Since other species don’t seem strictly relevant to the topic at hand, I was briefly wondering if it may be better to remove such information. However I don’t insist on this, as I can imagine circumstances where this info could be useful for other researchers.

Reply 4-6. For Japanese researchers of millipedes and soil biologists to know exact locations, we like to keep Figure 1 in the main text 1.

Line 528. Delete “are”

Reply 4-7. Done. Corrected copy: L537. Clean copy: P19 L528.

Line 538. At the beginning of the line, replace “brood” by “broods”.

Reply 4-8. Done. Corrected copy: P16 L543. Clean copy: P20 L534.

Line 543. Tables 1-3 may be of little interest to the average reader, so perhaps could be moved to the supplementary material.

Reply 4-9. Thanks for this comment. The reviewer 3 also suggested this move. Moved (Tables 1-3 → table S6-S8).

End of the reply letter

Appendix C

One-to-one response to reviewers

Associate Editor Comments to Author (Dr Jake Socha):

The reviewers have no further major comments and are satisfied with the revisions. Congratulations on the acceptance of this manuscript. However, please consider addressing reviewer 1's few comments to address language issues prior to final publication.

REPLY: Thanks a lot. We enclosed the English editing certificate of the previous revision.

Reviewer comments to Author:

Reviewer: 4

Comments to the Author(s)

I am satisfied with the revision, but have noticed some small linguistic errors in the revised text (see below).

REPLY: Thanks a lot.

L322: replace "apparently does fit" with "apparently do not fit"

REPLY: Thanks. Corrected. (Line 322 in the corrected ms.)

L324 replace "is the mistake of...days" with "may instead refer to the adjacent brood V, which was not known in those days"

REPLY: Thanks. Corrected. (Line 324 in the corrected ms.)

L325 replace "on 1943" with "in 1943"

REPLY: Thanks. Corrected. (Line 329 in the corrected ms.)

L327 replace "of the defeated" with "of the defeat in"

REPLY: Thanks. Corrected. (Line 331 in the corrected ms.)

L337 replace "Thus ... including" with "Thus, the 8-year periodicity is widely confirmed in all broods (except these two unconfirmed cases), including"

REPLY: Thanks. Corrected. (Line 333-334 in the corrected ms.)

Reviewer: 3

Comments to the Author(s)

The authors have made changes according to reviewers' suggestions. The revised manuscript would be acceptable.

REPLY: Thanks a lot.